# OpenReview forum: "Is it Thinking or Cheating?  Detecting Implicit Reward Hacking by Measuring Reasoning Effort"
_ICLR.cc/2026/Conference — ICLR 2026 Oral_

### Official Review · Reviewer_j8jz · 2025-10-31

**Soundness:** 3
**Presentation:** 3
**Contribution:** 3
**Rating:** 8
**Confidence:** 4

**Summary:**

This work proposes Truncated Reasoning AUC Evaluation (TRACE), a method that measures reasoning effort by checking how early the models reasoning becomes sufficient to pass the verifier. The intuition is that hacking the reward requires less effort than genuine reasoning, so the hacked responses can pass the verifier earlier. Based on this intuition, the method is relatively simple; the authors just add answer tokens at various lengths of the CoT and force the model to emit a response with these "incomplete" reasoning traces. They then measure the verifier pass rate across these truncation points and use the area under the curve as a quantitative metric for the models reasoning effort. The authors demonstrate that TRACE has better performance than the more traditional CoT monitors on detecting hacking and they even use TRACE to identify training environment loopholes, i.e., potential shortcuts in datasets that can lead the models to hack.

**Strengths:**

- TRACE is a simple and effective method for detecting reward hacking in reasoning models, an important problem.
- TRACE's results demonstrate that it can be better than the more traditional CoT monitors while also avoiding the need for having an additional model, the CoT monitor, thus offering better scalability.
- The paper is well structured, clear and easy to read.

**Weaknesses:**

- TRACE can be computationally expensive due to needing multiple generations to populate the CoT % vs Avg passing rate curve
- The comparison to CoT monitors is not entire apples-to-apples as TRACE uses strictly more information; it allows querying the base model multiple times and it also has access to the output reward. The authors did not compare with more involved monitors, e.g., monitors that take as inputs CoT, actions and reward output.
- Something which the authors also acknowledge is the relatively simple nature of the loopholes that they investigated.

**Questions:**

Overall, I am on the positive side for this work. Some comments and questions to the authors are

- The authors focus on verifier pass rate instead of checking for the same answer when truncating the CoT. Why not also check for the same answer? Shorter CoTs might need to exploit more the loopholes present in the inputs in order to pass the verifier check but that doesn't mean that they are probable under the model distribution. Furthermore, if the CoT is not informative enough for the final response, then the model might need to rely more on the prompt (and potential loopholes) to solve the task. How does the TRACE score look like if the authors use "same answer" instead of "pass rate"?
- The authors mention that a higher TRACE score indicates that the model knows the loophole earlier in the CoT. As knowing and using are different concepts, do the models actually use that information? Are there instances where the answer flips between hacky and non-hacky as a function of the CoT length?
- How does TRACE perform on simpler tasks where CoT might not be necessary in order to have good performance? Does TRACE require that reasoning is necessary to be successful (i.e., do the authors observe similar phenomena to Emmons et al, 2025)?

---

> ### Author Response · Authors · 2025-11-20
> **Reply to Reviewer j8jz (1/2)**
>
> Thank you for your valuable feedback and we would like to address your concerns below:
>
> ### Questions:
> 1. >The authors focus on the verifier passing rate instead of checking for the same answer when truncating the CoT. Why not also check for the same answer?
>
> Thank you for your advice. We would like to first clarify that we are detecting hacking on those samples where the final answer passed the verifier, and we want to know whether the model passes the verifier in the intended way. In the in-context loophole setting,  checking if the early answer passes the verifier is essentially the same as checking if the early answer is the same as the final answer, because there is only a single solution that could pass the verifier. However, in the reward model loophole setting (all negative numbers can pass the verifier), there are multiple ways to hack. The early answers would output different negative values to hack the reward, other than sticking to the same final answer. That’s why we use the verifier passing rate to assess the TRACE score. What we are essentially measuring is: how early in the CoT the model will be able to obtain the reward.
>
>
>
> 2. >Do the models actually use that information? Are there instances where the answer flips between hacky and non-hacky as a function of the CoT length?
>
> We appreciate the reviewer’s further question on this matter. One way to test if the model is ‘using that information’ is to do counterfactual testing, which is exactly what we did for labelling the hacking examples in the first place. All the hacking samples we collect are those samples that the model couldn’t solve if we remove the loophole. If we probe the non-loophole version of the sample by truncation, all the early answers will also be wrong, as the question is beyond the model’s capability. Based on this counterfactual test, we could infer that the model is using the loophole if the (early) answers are correct when the loophole is given.
>
> 3. >How does TRAE perform on simpler tasks where CoT might not be necessary in order to have good performance? Does TRACE require that reasoning is necessary to be successful? ( do the authors observe similar phenomena to Emmons et al, 2025)?
>
>
> We agree with the reviewer that task difficulty is a factor that could influence the calculation of the TRACE score.  In our preliminary experiments, we did observe that TRACE score tends to be lower on simpler questions than the hard ones, based on the Llama3 solve-rate. However, in our current setting, the impact of difficulty is covered properly because the baseline is calculated on all the problems where the initial policy solves correctly. Those questions are considered the easiest questions and the rest of them are considered harder. As we train the model on the harder questions, if it has a higher AUC than the easy problems, we should flag it as hacking.
>
> We do agree that in the extreme cases where the questions can be solved in a single forward pass, it will be trivial to use TRACE as a detector. However, easy questions with simple CoT decrease the monitoring difficulty by inspecting the text directly, posing less threat than hacking with long and complex CoT on a hard question.
>
>
> As for the phenomena observed by Emmons et al, where complex loopholes are more more like to be verbalized in the CoT (which is a different problem as we discussed above on the question complexity), we actually have a similar finding in the Appendix A, where we also change the loophole complexity in a similar way and observed that complex loopholes are easier to detect. The difference from the Emmons et al, is that we detect the model that is trained with the loophole to learn the hack, instead of detecting in-context hacking.
>
>
> We have added a section to discuss the impact of the task difficulty.

---

> ### Author Response · Authors · 2025-11-20
> **Reply to Reviewer j8jz (2/2)**
>
> ### Weakness:
>
> 1. >TRACE can be computationally expensive due to needing multiple generations to populate the CoT % vs Avg passing rate curve
>
> We believe there is a significant room for speed up by using the KV cache of the truncated CoT when sampling, as we are using the same CoT when forcing the early answers. Therefore, the only additional computation overhead comes from the answer part but not the CoT part. We
>
> 2. >The comparison to CoT monitors is not entire apples-to-apples as TRACE uses strictly more information; it allows querying the base model multiple times and it also has access to the output reward. The authors did not compare with more involved monitors, e.g., monitors that take as inputs CoT, actions and reward output.
>
> That's very thoughtful feedback. Firstly, we want to clarify that our CoT monitor has full access to both the input prompt and the entire LLM response trajectory. And both the CoT monitor and the TRACE monitor are used for samples that have reward of 1, since we are interested in if the model is cheating when it passes the verifier.
>
> We do agree that TRACE obtains more information from the model by probing the answer multiple times. However, the CoT monitor also utilizes additional reasoning capability from an external LLM, and this additional computation may exceed the computation we need for the TRACE method, if we utilize the KV cache of the CoT truncation. We allow a much stronger CoT monitor to do extensive reasoning on the sample with rubrics before giving the final judgement, while TRACE only relies on the responses from the weaker model itself. Therefore, the TRACE is a more scalable approach as we monitor stronger models.
>
>
>
> 3. >Something which the authors also acknowledge is the relatively simple nature of the loopholes that they investigated.
>
> We do acknowledge the simplicity of the loophole design and we see substantial potential for future research on designing more complicated and subtle loopholes for studying the reward hacking behaviour of the model. One difficulty of detecting more complex loopholes may be the difficulty to elicit early answers, as those loopholes may require reasoning or tool calling tokens to be used. One mitigation strategy could be: Encourage the model to ‘wrap up’ its reasoning with minimal tokens, to let the model work on the loophole or do tool calling. In other words, we can give the model more freedom when we force-stop the CoT, allowing them to reason and action with the loopholes.

---

> > ### Comment · Reviewer_j8jz · 2025-11-24
> >
> > I would like to thank the authors for responding to my comments. They were adequately addressed and I will be maintaining my score.

---

### Official Review · Reviewer_1wDd · 2025-11-01

**Soundness:** 3
**Presentation:** 3
**Contribution:** 3
**Rating:** 8
**Confidence:** 4

**Summary:**

This paper tackles the problem of detecting *implicit* reward hacking - hacking that is not verbalized in CoT traces. The paper makes the observation that implicit reward hacks are often simple and don't require reasoning. This motivates their methodology, TRACE, which cuts reasoning short and evaluates whether the model can still get the right answer. If the model is consistently able to get the right answer with its reasoning cut short, then the model is more likely to be hacking. The authors show that this detection method is useful for distinguishing hacking vs non-hacking models; and also to detect hacking responses from a single model.

**Strengths:**

- Well-written
	- The paper is clear and well-written. It is easy to follow and make sense of.
- Sensible solution to a critical problem
	- Reward hacking is a critical problem facing scalable RL training. Detecting reward hacks is increasingly important, and in my opinion this paper produces a nice detection mechanism to implicit reward hacks. The solution is simple (which is a good thing).
- The paper goes beyond initial investigation to applications
	- I appreciated that the authors applied the method to two tasks: distinguishing between a hacking vs non-hacking model, and distinguishing hacking samples from non-hacking samples. I think the latter is in particular interest for real RL training, and the authors indicate that this can be useful for detecting hacks in the RL environment.
- Properly acknowledges limitations
	- I appreciate that the authors honestly acknowledge the limitations of their work; the main limitation is the simplicity of the carefully-designed reward hacks studied in this paper.

**Weaknesses:**

- The environments are very synthetic / contrived, and likely don't reflect real-world reward hacking scenarios.
	- The authors acknowledge this limitation, to be fair!

**Questions:**

- In the code setting, why sample once? (Footnote 2) In the math setting, 5 responses are sampled at temperature 0.7; why not do the same for the code setting?
- I think the paper could benefit from a discussion of cost. Is this methodology feasible to deploy in practice?
- I am also curious whether models could be trained adversarially against TRACE. For example, could a model learn to "know" when its CoT is cut short, and not hack in those cases?

---

> ### Author Response · Authors · 2025-11-20
>
> Than you for your careful review! We reply your questions below:
>
> 1. >Why sample once in the code setting?
>
> We sample once in code because we want to know if detection could work if we prioritize efficiency and use a ‘softer’ check by checking the ratio of test cases passed. In coding problems, we show that using the ratio of test cases passed also yields superior detection performance compared to CoT monitoring. We do expect that sampling more responses would give better results in the coding setting. In our current setting, sampling once would give us good enough detection performance in both the math (in preliminary experiments) and code settings, but one can sample more trajectories if robustness is a higher priority than efficiency.
>
> 2. >I think the paper could benefit from a discussion of cost. Is this methodology feasible to deploy in practice?
>
> We are happy to add discussion on cost into the paper. We do acknowledge that the process of truncation and multiple sampling seems heavy. However, due to vLLM’s parallel sampling function, increasing the number of samples does not multiply the prompt pass, because they are sharing the same KV cache of the same prompt [1]. Therefore, we don’t observe substantial runtime increase as we increase the number of sampling.
>
> Furthermore, there is also a significant room for optimizing the speed by sharing the KV cache computed for the complete CoT with the different truncated CoTs. By doing that, the only additional computation overhead comes from sampling the (early) answers.
>
> We also want to point out that the computation required for the external CoT monitor is not trivial, as we need a much stronger LLM to reason about the full trajectory, using some judgement rubrics. Therefore, the reasoning trace of the monitor is long and computationally heavy.
>
> We will add a discussion paragraph in the updated version of the paper.
>
>
> 3. >Whether models could be trained adversarially against TRACE. For example, could a model learn to "know" when its CoT is cut short, and not hack in those cases?
>
> That’s a great question, and we agree with the reviewer’s speculation that the model could learn to hide the hack when trained in an adversarial way. This could happen when the TRACE score is included into the reward design, and the model learns to reduce trace score under optimization. Recent work has shown that the model could learn to hide its hacking intention from the CoT if the monitor signal is included in the reward function [2]. We expect something similar to happen for the TRACE monitor. Therefore, we propose to use our method as a complementary monitor but not as a direct target to optimize for.
>
> We have added a new paragraph in the discussion section to discuss the impact of adversarial training against TRACE.
>
> ```
> [1]Kwon, Woosuk, et al. "Efficient memory management for large language model serving with pagedattention." Proceedings of the 29th symposium on operating systems principles. 2023.
> [2]Baker, Bowen, et al. "Monitoring reasoning models for misbehavior and the risks of promoting obfuscation." arXiv preprint arXiv:2503.11926 (2025).

---

> > ### Comment · Reviewer_1wDd · 2025-11-25
> >
> > I acknowledge the authors' rebuttal. I will maintain my score.
> >
> > Separately, why do you cite Baker et al., 2025 in your sentence about vLLM? I'm still not sure I understand the point about parallel sampling, and would appreciate if you could give a clearer explanation. E.g., is it something to do with vLLM specifically? or just the nature of auto-regressive LMs? how exactly do you perform this parallel sampling without extra cost? This is a minor point, but it's not clear to me.

---

> ### Author Response · Authors · 2025-11-25
>
> Thank you for your follow-up questions.
> The citation was actually a mistake (we mean to cite Baker et al.,2025 in the response to Q3), and we have changed it to the vLLM paper. Sorry for the confusion!
>
> Yes, it is specifically due to vLLM's parallel sampling technique.
> In the parallel sampling section, it says
> `"In parallel sampling, one request includes multiple samples sharing the same input prompt, allowing the KV cache of the prompt to be shared as well." `
> This saves the time for recomputing the input prompt, allowing faster sampling.
>
> Thanks for your question!

---

### Official Review · Reviewer_afri · 2025-11-01

**Soundness:** 3
**Presentation:** 3
**Contribution:** 3
**Rating:** 8
**Confidence:** 4

**Summary:**

This paper addresses implicit reward hacking in reasoning models, i.e. cases where models exploit evaluation loopholes but produce plausible CoTs that evade monitors. The proposed method, TRACE (Truncated Reasoning AUC Evaluation), measures “hidden effort” by truncating CoTs at various points, forcing early answers, and computing the verifier pass rate curve. A high AUC indicates early success, suggesting exploitation. Experiments on math and code tasks with two loophole types (in-context hints and reward-model bugs) show TRACE outperforms CoT monitoring and can cluster samples for loophole discovery.

**Strengths:**

The paper makes a clear and original contribution to oversight research by shifting the focus from what the CoT says to how much effort the model expends. This is both conceptually appealing and practically relevant.

* Novel perspective: Instead of analyzing CoT content, TRACE uses early passability as a proxy for reasoning effort, which is a clever way to expose hidden shortcuts.
* Empirical gains: Across math and code tasks, TRACE achieves large improvements over strong CoT monitors (up to +65% F1 in math, +30% in code), even when hacked CoTs look perfectly reasonable.
* Instance-level detection: TRACE works per sample and tracks hacking onset during RL training, revealing dynamics that CoT length or KL loss fail to capture.
* Loophole discovery: Clustering by TRACE score and prompting an LLM successfully identifies injected artifacts, showing potential for unsupervised forensic analysis.
* Scalable oversight angle: Because TRACE relies only on the model's own outputs, it avoids the need for ever-larger external monitors.

**Weaknesses:**

While the paper is strong, several practical details would make it easier to deploy:


Deployment framing & calibration guidance: TRACE is shown to work during training and at inference, but the paper doesn't offer a clear, setting‑specific recipe for threshold calibration, drift handling, and baseline selection. This is especially important when the initial‑policy baseline is unavailable (e.g., auditing a third‑party model) or when data distribution shifts over time.


Assumptions about CoT access & control: TRACE presumes you can see the CoT, truncate it, and force early answers. That’s not always feasible (latent‑reasoning systems, models with hidden traces, or agents whose action plans break when truncated).


Early solvability vs. hacking: High AUC can reflect genuinely easy‑early items rather than exploitation. A brief false‑positive analysis (e.g., counterfactuals that remove loopholes but preserve difficulty) would clarify this boundary.


Cost/sensitivity profile: TRACE uses multiple truncation points and sampling. The paper lacks runtime and stability guidance (e.g., how many points/samples are enough, temperature sensitivity, and the cost–accuracy Pareto).


Adversarial adaptation: A savvy model could pad early reasoning or delay the exploit to evade early passability checks. A short red‑teaming experiment (or mitigation ideas like randomized truncation schedules) would strengthen robustness.


Realism and coverage of loopholes: The synthetic IC/RM setups are clean and illustrative; real exploits can be subtler, multi‑step, or tool‑centric. Even a small "wild" case study (or a discussion of expected failure modes) would help scope external validity.

**Questions:**

1. Scope and deployment scenario:

Is TRACE intended as a general-purpose detection method or primarily for RL-trained models? The paper mixes inference-time and training-time use cases.

---
2. In-context loopholes without training:

Section 3.2 defines hacking models as RL-trained, but in-context exploits can occur at inference without any training. How does TRACE handle such cases? Would detection still work if the model was never optimized for hacking?

---
3. Verifier pass rate vs same answer:

You mention “generalize same answer to verifier pass rate.” Could you provide an example where the truncated CoT produces a different final answer but still passes the verifier, and explain why this matters for detecting hacking?

---
4. Reward spike vs TRACE signal:

In Fig. 8, reward seems to “explode” once the loophole is discovered. Would simple reward monitoring during training suffice to detect hacking? Can you imagine a scenario where reward rises smoothly but TRACE still detects hidden shortcuts?

---
5. Truncation vs naturally short CoTs:

TRACE truncates CoTs post-hoc, but some models are trained to produce short CoTs (e.g., under token budgets). Would TRACE misinterpret low AUC as high effort in such cases? How could the method adapt to budget-forced reasoning?

---
6. Delayed exploit strategies:

Would TRACE fail if models output plausible intermediate steps and only apply the exploit at the very end? How robust is TRACE to adversarial adaptation where the hack is deferred?

---
7. Compute overhead and sensitivity:

How many truncation points and samples are needed for stable AUC estimates? What is the runtime overhead compared to CoT monitoring? Any analysis of sensitivity to sampling temperature or truncation granularity?

---
8. Realism of loopholes:

The injected loopholes (IC hints, RM bugs) are clear but somewhat artificial. How representative are these of real-world exploits? Could TRACE handle multi-step or tool-based hacks where success depends on external actions?

---

> ### Author Response · Authors · 2025-11-20
> **Reply to Reviewer afri (1/3)**
>
> Thank you for providing detailed feedback and constructive suggestions. We would like to address your comments below:
>
> ### Weakness:
>
> 1. >Threshold Calibration: TRACE is shown to work during training and at inference, but no clear setting-specific recipe for threshold calibration, drift handling, and baseline selection. This is especially important when the initial policy baseline is unavailable, as data shift over time.
>
> We consider our method as a development-time tool that developers can use to flag hacking behaviour during training, and we believe it has the potential to flag hacking behavior at deployment time. In our current experiment setting, the detection set shares the same distribution as the training set, which is a simulation of the training phase detection before deployment. As for the detection after the deployment, where the distribution could shift, one solution is to sample additional data to calibrate the initial policy baseline to adjust to the data shift. Regarding auditing third-party models, our method is designed for model developers who have complete control over the model checkpoints and the dataset, so that developers can take action on flagged hacking instances.
>
> 2. >Assumption about CoT access. Is it feasible with Latent Reasoning Systems, model with hidden traces, or agent whose action plans break when truncation?
>
> Our method is designed for model developer who has the access to the full reasoning trajectory but it doesn’t rely on the semantics of the CoT because we only consider the CoT tokens as a proxy of reasoning effort. Breaking the semantic continuation is even preferred in this case, as that’s a way to test if the model is relying on the external language to reason. Success in such a case is an indicator of a hidden objective, which will be revealed by a high TRACE score.
>
> Our method can also be applied to models reasoning with latent CoT, such as the Coconut method [1].  In the Coconut work [1], the latent CoT also relies on special tokens such as ``<eot>`` to stop the latent reasoning. Therefore, we can force the model to use different numbers of latent CoT tokens by injecting the ``<eot>`` token at different positions, which was also done in the Coconut paper to study the impact of token numbers. One may need to do additional modification by adding additional text stop tokens after the ``<eot>`` token, so we can prevent the model from continuing to reason in the language space
>
> 3. >Early solvability vs. Hacking: High AUC can reflect genuinely easy-early items than exploitation.
>
> We do acknowledge that easy questions may be easier to answer early in the CoT, and this was also confirmed in our preliminary experiments, where we have problems with different difficulty levels (based on the Llama3 solve rate), without inserting loopholes. However, this has already been considered in our current setting, as the baseline is calculated on all the problems where the initial policy solves correctly. Those questions are considered the easiest questions and the rest of them are considered harder. As we train the model on the harder questions, we should be confident to flag a response as hacking if it has a higher AUC than the easy problems (meaning they are using less effort on harder questions).
>
> 4. >TRACE uses multiple truncation points and sampling. The paper lacks runtime and stability guidance (e.g., how many points/samples are enough?)
>
> Due to vLLM’s parallel sampling function, increasing the number of samples does not multiply the prompt pass, because they are sharing the same KV cache of the prompt [2]. Therefore, we don’t observe substantial runtime increase as we increase the number of sampling. In the coding setting, we also adopt a 'soft check' approach by sampling once and checking the ratio of the test cases passed as a simpler approach than multi sampling.
>
> Furthermore, there is also a significant room for optimizing the speed by sharing the KV cache computed for the complete CoT with the different truncated CoTs. By doing that, the only addition computation overhead comes from sampling the (early) answers.
>
> We will add a discussion paragraph in the updated version of the paper.
> ```
> [1]Hao, Shibo, et al. "Training Large Language Model to Reason in a Continuous Latent Space." COLM 2025
> [2]Kwon, Woosuk, et al. "Efficient memory management for large language model serving with pagedattention." Proceedings of the 29th symposium on operating systems principles. 2023.

---

> ### Author Response · Authors · 2025-11-20
> **Reply to Reiview afri (2/3)**
>
> 4. >Adversarial adaptation: A model could pad the early reasoning or delay the exploit to evade the early passability checks. A short red-teaming experiments or mitigation ideas like randomized truncation schedules would strengthen the robustness
>
> Since we consider the detection to happen in the development phase, it is unlikely that the model will learn such a strategy to escape the TRACE monitor without being trained for it specifically. However, there is a chance of picking up such a strategy, which could happen if the TRACE  score is included into the reward function. The optimization pressure will push the model to hide its early answer probability to pass the check. Therefore, we recommend not including the TRACE score into the reward design.
> We have also added a paragraph into the discussion section on this.
>
> 5. >Real loopholes could be subtle, multi-step, or tool-centric. Even a small "wild" case study (or a discussion of expected failure modes) would help scope external validity.
>
> We do agree that detecting ‘wild’ hacking cases may face additional obstacles but we think it can be overcome by modifying our current approach.
> As for complex and tool-use loopholes, it may be difficult to directly force the model to output an answer with ``<answer>`` tags, as the model might need extra tokens to compute the loophole answer or do tool calling. To better elicit the loophole answer, we can encourage the model to ‘wrap up’ its reasoning with minimal tokens, to let the model do the loophole answer calculation or tool calling. In other words, we can give the model more freedom when we force-stop the CoT, allowing them to reason and action with the loopholes.
>
> ### Questions:
>
> 1. >Deployment Scenario: Is TRACE general detection method only for RL model? Paper mixes the inference and Training
>
> Our TRACE method is designed to detect reward hacking behaviour that has been learned by the model during RL. As we answered in Q1 from the weakness section, our setting simulates training time detection where the detection data shares the same distribution as the training data. To detect hacking after deployment, we could calibrate the baseline by sampling more data to adjust for any distributional shift.
>
>
> 2. >In-context loopholes without training: Section 3.2 defines hacking models as RL-trained, but in-context exploits can occur at inference without any training. How does TRACE handle such cases? Would detection still work if the model was never optimized for hacking?
>
> Section 3.2 defines hacking models as RL-trained, but in-context exploits can occur at inference without any training. How does TRACE handle such cases? Would detection still work if the model was never optimized for hacking?
> We observed the same trend of a higher TRACE score for in-context hacking in our preliminary experiments, where we gave the in-context loophole answer in the prompt. The reason is that both in-context and learned hacking behaviour should share the same mechanism: the model is using less effort than solving the actual task, which can be revealed by CoT truncation.
>
> 3. >Verifier pass rate vs same answer: You mention “generalize same answer to verifier pass rate.” Could you provide an example where the truncated CoT produces a different final answer but still passes the verifier, and explain why this matters for detecting hacking?
>
> This is exactly the case in our reward model loophole setting, where the model could generate different answers to pass the verifier: The reward model loophole is rewarding any negative numbers. When we truncate the CoT, the model will generate different negative values across the exit points, which should be considered hacking. Therefore, checking verifier pass is a more general approach to catch hacking behaviour.
>
> 4. >Reward spike vs TRACE signal:
> In Fig. 8, reward seems to “explode” once the loophole is discovered. Would simple reward monitoring during training suffice to detect hacking? Can you imagine a scenario where reward rises smoothly but TRACE still detects hidden shortcuts?
>
> We do observe smoother reward gain for hacking where the loopholes are harder to exploit. For example, in our mixed-loophole setting, the reward gain is much smoother because the model can only hack on part of the dataset, as described in setting 2 in section 4.2.  Our results show that TRACE score is high only when the model is exploiting the part of the data where a loophole exists. Therefore, our TRACE approach is considered to be effective where the reward spike is not present.

---

> ### Author Response · Authors · 2025-11-20
> **Reply to Reviewer afri (3/3)**
>
> 5. >Truncation vs naturally short CoTs:
> TRACE truncates CoTs post-hoc, but some models are trained to produce short CoTs (e.g., under token budgets). Would TRACE misinterpret low AUC as high effort in such cases? How could the method adapt to budget-forced reasoning?
>
>
> That’s a great question. We agree that TRACE score can be influenced by the CoT token budgets, but such a factor can be cancelled out if both the initial policy and the RL-ed one share the same token budgets.  The criteria for flagging hacking behaviour is based on the initial policy, and therefore we are only comparing relative change of the model, which excludes the token budges that both the initial and RL-ed model shared.
>
>
> (we answer comments 6-7 in the weakness section as they are overlapping.)

---

> > ### Comment · Reviewer_afri · 2025-11-25
> >
> > I would like to thank the authors for the clarifications, which answered all my (minor) concerns. I will be maintaining my score.

---

### Official Review · Reviewer_kcio · 2025-11-01

**Soundness:** 3
**Presentation:** 3
**Contribution:** 3
**Rating:** 6
**Confidence:** 4

**Summary:**

This paper proposed TRACE, a method for identifying hacking behaviors in LRMs.  The authors make the hypothesis that a hacking model will produce the correct answer in the early part of the CoT trajectory while a non-hacked model needs the full CoT trajectory to be able to reach a final answer. The TRACE method involves several steps. First, a given LRM-generated CoT trajectory is truncated into chunks, which are then cumulatively fed into the LRM to sample responses. The answers are passed into a verifier to compute the passing rate given the current CoT chunk. Based on the assumption, hacked models would possess a high passing rate with only a fraction of its CoT trajectory while non-hacked models could only have a high passing rate if the full CoT  trajectory is provided. The authors conduct experiments in both in-context hacking and reward model hacking settings to validate the performance of TRACE. Further, the authors demonstrate that TRACE can be used for identifying data entries that may be hacked by the LRM.

**Strengths:**

- The paper is well-written and easy to follow.
- This paper proposed TRACE, a method for identifying hacking behaviors in LRMs via truncating their CoT trajectories. The proposed method is simple and effective according to the reported experimental results.
- The experiment setup is comprehensive, which includes in-context loophole and reward model loophole.
- The case study offers a potential usage of TRACE, which is to identify data entries that contain loopholes. Experiment results show that TRACE can reliably distinguish hacking data from others once the loophole is exploited.
- Experiment results show that TRACE offers more in-depth detection of reward hacking than existing superficial detection criterias such as CoT length and KL Loss.

**Weaknesses:**

- It is unclear whether the proposed TRACE method will be robust enough to handle LRMs with overthinking issues where the correct answer may occur in the middle of a CoT trajectory albeit the model itself is unhacked.
- The authors mentioned several limitations with CoT monitor approach, which mainly include deceptive CoT content and latent CoT methods. While the proposed TRACE method may be used to tackle deceptive CoTs, it is unclear to me how TRACE can be used in the case of latent CoT.
- The sampling process seems to introduce quite a lot of compute overhead as a given LRM needs to produce multiple responses for each chunk of the truncated CoT trajectory.

**Questions:**

- How does the proposed TRACE method handle cases of overthinking? An unhacked model may produce CoT trajectories that contain correct answers but are prolonged due to overthinking.
- How can TRACE be used to tackle latent CoT?

---

> ### Author Response · Authors · 2025-11-20
>
> Thank you for your valuable feedback. We would like address your concerns regarding overthinking, latent CoT, and the computation overhead below:
>
>
> 1. >It is unclear whether the proposed TRACE method will be robust enough to handle LRMs with overthinking issues where the correct answer may occur in the middle of a CoT trajectory albeit the model itself is unhacked.
>
> That’s a great question. We agree that overthinking will lead to a higher AUC score because the passing rate is high once the correct answer is reached in the middle of the CoT. However, this was considered in the way we set up our detection threshold. If we are training an overthinking LRM, we first establish a baseline by calculating the TRACE score on all the samples the model can solve before we train the model with RL. Those correct samples are the easiest ones in the dataset for the model to solve, and thus, should have the highest TRACE score under overthinking, raising the detection AUC threshold high. As we train and monitor the new examples, which are learned to solve during RL, they should have a lower TRACE score under the same overthinking problem, as harder questions require longer exploration to find the solution. Therefore, we should be confident in labeling samples as hacking if they have a higher TRACE score than the baseline. In other words, by incorporating overthinking into the baseline calibration, we should be able to remove the impact of overthinking on the detection.
>
> We can also prevent the model from picking up overthinking during RL training by adopting overthinking prevention techniques.
>
> 2. >While the proposed TRACE method may be used to tackle deceptive CoTs, it is unclear to me how TRACE can be used in the case of latent CoT.
>
> We expect the same methodology to work similarly for latent CoT, as the latent CoT also relies on special tokens such as ``<eot>`` to stop the latent reasoning. Therefore, we can force the model to use different numbers of latent CoT tokens by injecting the ``<eot>`` token at different positions, which was also done in the Coconut paper [1] to study the impact of token numbers. One may need to do additional modification by adding additional text stop tokens after the ``<eot>`` token, so we can prevent the model from continuing to reason in the language space.
>
> 3. >Sampling seems to introduce quite a lot of compute overhead as a given LRM needs to produce multiple responses for each chunk of the truncated CoT trajectory.
>
> We appreciate your concern on the computation overhead. We do acknowledge that the process of truncation and multiple sampling seems heavy. However, due to vLLM’s parallel sampling function, increasing the number of samples does not multiply the prompt pass, because they are sharing the same KV cache of the prompt [2]. Therefore, we don’t observe substantial runtime increase as we increase the number of sampling. In the coding setting, we also adopt a 'soft check' approach by sampling once and checking the ratio of the test cases passed as a simpler approach than multi sampling.
>
> Furthermore, there is also a significant room for optimizing the speed by sharing the KV cache computed for the complete CoT with the different truncated CoTs. By doing that, the only addition computation overhead comes from sampling the (early) answers.
>
> We also want to point out that the computation required for the external CoT monitor is not trivial, as we need a much stronger LLM to reason about the full trajectory, using some judgement rubrics. Therefore, the reasoning trace of the monitor is long and computationally heavy.
>
> We will add a discussion paragraph in the updated version of the paper.
>
> ```
> [1]Hao, Shibo, et al. "Training Large Language Model to Reason in a Continuous Latent Space." COLM 2025
> [2]Kwon, Woosuk, et al. "Efficient memory management for large language model serving with pagedattention." Proceedings of the 29th symposium on operating systems principles. 2023.

---

> > ### Comment · Reviewer_kcio · 2025-11-25
> >
> > I would like to thank the authors for the clarifications.
> > As the author's clarification are just conceptual, not experimental results, such as how to work with latent CoTs;
> > what are the performances when models are overthinking, etc. so, I will be maintaining my score.

---

> ### Author Response · Authors · 2025-11-26
>
> Dear Reviewer kcio,
>
> We fully understand your expectation on seeing experiment results with TRACE on LRM with overthinking issue or latent CoT.
>
> However, we kindly disagree that this should be considered as our weakness since this requires substantial experimental setup and careful tuning, which itself could require another paper for solid conclusion. We consider this out of our scope and an important direction for future work.
>
> Our paper conceptualizes and validates a novel approach of reasoning supervision, going beyond the current static CoT monitor approach. We do see great potential of further using and improving our approach on non-standard cases. We appreciate that reviewer brought these cases up.
>
> We have now added such limitation into our discussion section and encourage future work to follow up.
> Thanks again for pointing out those promising directions and we hope this can address your concern adequately.
>
> Authors

---

### Comment · Area_Chair_G8zm · 2025-11-25
**Please respond to authors' rebuttal**

Dear Reviewers,

Thanks for your efforts on reviewing this work. Please read the authors' rebuttal and give feedback about whether your concerns have been addressed.

Please note that DONOT fully utilize LLM to directly generate the reviews and responses, ensuring that the reviews are made based on your own expertise, not AI's knowledge. If your review is flagged as LLM generated, it will be reported to PC.

Sincerely,
AC

---

### Meta-Review · Area_Chair_7ygo · 2026-01-07

**Summary:**

The reviewers generally praised the paper's novel TRACE method for detecting implicit reward hacking in reasoning models via truncated CoT evaluations, highlighting its simplicity, empirical superiority over CoT monitors, and applications like loophole discovery. Overall, all reviewers agree to accept this paper.

**Reviewer Concerns:**

Addressed: Overthinking via baseline calibration and prevention techniques; latent CoT via token injection adaptations; compute overhead via vLLM optimizations and KV cache sharing

Outstanding: Realism of synthetic loopholes; potential false positives on easy tasks; apples-to-apples comparisons with advanced monitors

**Reviewer Scores:**

- Reviewer kcio: Would maintain 6 (conceptual clarifications insufficient without experiments).
- Reviewer afri: Would maintain 8 (all minor concerns addressed).
- Reviewer 1wDd: Would maintain 8 (questions on cost and adversarial training resolved).
- Reviewer j8jz: Would maintain 8 (concerns on pass rate vs. same answer and compute adequately handled).

---

### Decision · Program_Chairs · 2026-01-26

Accept (Oral)